# Bactofection, Bacterial-Mediated Vaccination, and Cancer Therapy: Current Applications and Future Perspectives

**DOI:** 10.3390/vaccines12090968

**Published:** 2024-08-27

**Authors:** Francisco Israel Renteria-Flores, Mariel García-Chagollán, Luis Felipe Jave-Suárez

**Affiliations:** 1Institute of Research in Biomedical Sciences, University Center of Health Sciences (CUCS), University of Guadalajara, Guadalajara 44340, Jalisco, Mexico; renteriaf33@gmail.com (F.I.R.-F.); chagollan@academicos.udg.mx (M.G.-C.); 2Division of Immunology, Biomedical Research Centre of the West, Mexican Social Security Institute, Guadalajara 44340, Jalisco, Mexico

**Keywords:** bactofection, bacterial based vaccination, bacterial cancer, gene therapy

## Abstract

From the first report in 1891 by Dr. Coley of the effective treatment of tumors in 1000 patients with Streptococcus and the first successful use of bacterial vectors for transferring therapeutic genes in 1980 by Dr. Schnaffer, bactofection has been shown to be a promising strategy in the fields of vaccination, gene therapy, and cancer therapy. This review describes the general theory of bactofection and its advantages, disadvantages, challenges, and expectations, compiling the most notable advances in 14 vaccination studies, 27 cancer therapy studies, and 13 clinical trials. It also describes the current scope of bactofection and promising results. The extensive knowledge of *Salmonella* biology, as well as the multiple adequacies of the Ty21a vaccination platform, has allowed notable developments worldwide that have mainly been reflected in therapeutic efforts against cancer. In this regard, we strongly recommend the creation of a recombinant Ty21a model that constitutively expresses the GtgE protease from *S. typhimurium*, allowing this vector to be used in animal trials, thus enhancing the likelihood of favorable results that could quickly transition to clinical trials. From the current perspective, it is necessary to explore a greater diversity of bacterial vectors and find the best combination of implemented attenuations, generating personalized models that guarantee the maximum effectiveness in cancer therapy and vaccination.

## 1. Introduction

Bactofection is a technique that uses bacteria-mediated transfer of DNA, RNA, or even translated proteins into a wide variety of mammalian cells (phagocytic or no phagocytic) [1]. This type of gene transfer relies on the ability of certain bacteria to invade eukaryotic cells and deliver their cargo material; therefore, it is a powerful tool to express exogenous genes that could encode regulatory RNAs or heterologous proteins (protein antigens, hormones, toxins, or enzymes) [1,2,3]. Bactofection has been used in a variety of research applications, including gene therapy, drug discovery, and basic research on gene function. The technique offers several advantages over other gene transfer methods, such as high efficiency, low toxicity, and the ability to deliver large DNA fragments [4,5].

The first recorded use of bacterial vectors to transfer genes into a target organism or tissue for therapeutic or research purposes dates to 1980, when Schnaffer tested a recombinant *Escherichia coli* harboring a plasmid with three tandem inserts of simian virus 40 DNA complete genome that was able to infect CV-1 monkey cells. The gene transfer was resistant to DNase I digestion, demonstrating this does not occur via free DNA but most likely via uptake of the whole bacteria. These results indicated for the first time that DNA transfer from bacteria to mammalian cells is a general phenomenon [2].

Live attenuated transformed bacterial strains can be used as vectors to deliver genes cloned into plasmids to eukaryotic cells or tissues. These genes can be expressed to produce a therapeutic product or to induce an immune response against a certain kind of antigen whose origin is infectious or oncogenic (Figure 1).

Overall, bactofection is a promising approach for gene delivery. Its potential applications include gene therapy, vaccine development, and genetic engineering.

The basic principle of bactofection relies on three main mechanisms. First, it takes advantage of natural bacterial invasion mechanisms to target specific tissues; this process is mediated by the genes contained in the pathogenicity islands of the bacterial genome itself. Second, it ensures the bacterial vector’s escape from the phagocytic vacuole of the antigen-presenting cells (APCs) by including specific virulence factors (e.g., listeriolysin O endotoxin from *Listeria monocytogenes*). Finally, the release of the cargo must be ensured, which is achieved by the induction of bacterial vector lysis through inducible autolytic mechanisms, metabolic attenuation (auxotrophy), or treatment with antibiotics. The release of cargo nucleic acids leads to the expression of the encoded protein (Figure 2). Many variations of this concept may appear according to the chosen bacterial vector and the structure of the implemented DNA construct (plasmid vector) [6,7,8,9].

A promising benefit of using bacterial vectors to deliver foreign DNA into eukaryotic cells, especially in vaccination or cancer therapy, is their capacity to act as a potent adjuvant, mainly because of their cell wall components (lipopolysaccharide, flagellin, peptidoglycan, teichoic acids, porins, mycolic acid, and mannose-rich glycans) that by their nature as pathogen-associated molecular patterns (PAMPs) induce the activation of pattern recognition receptors (PRRs) on immune cells, triggering potent activation, and consequently, leading to more efficient adaptive immune responses against the vector-encoded heterologous antigen. An excellent illustration of this was the proof that a bacterial-based oral vaccine platform can achieve long-lasting mucosal immunity, eliciting adequate B and T cell memory responses with long-lasting conferred immunity and producing fewer side effects than other vaccination platforms [4,5].

Likewise, bactofection has been quite effective for tumor treatment. In a study using live attenuated *Salmonella typhimurium* as a vehicle for oral gene therapy against two different murine tumors, transformed *S. typhimurium* harboring eukaryotic expression vectors containing genes for interleukin-12 (IL-12), granulocyte/macrophage colony-stimulating factor (GM-CSF), and green fluorescent protein (GFP), contributed to the increase of cytokines production in murine sera. This led to an increase in cytotoxic CD8+ lymphocytes and a significant survival rate improvement [10].

Vaccination or cancer therapy is not the only goal that can be accomplished with bactofection; bacterial vectors can also be used to deliver eukaryotic expression plasmids to introduce relevant transgenes into somatic tissues to treat or prevent diseases that result from genetic disorders. Of note are the experiments of Krusch and colleagues, who used a transformed and partially attenuated (reduced toxigenic activity of hemolysin) *Listeria monocytogenes* vector to deliver a cDNA encoding the human cystic fibrosis transmembrane conductance regulator (CFTR) to CHO-K1 cells, using antibiotics to diminish bacterial survival. Microscopic and biochemical methods effectively detected both transient and stable expression of the CFTR transgene, suggesting that *L. monocytogenes* facilitated the transfer of functional CFTR to CHO-K1 cells and indicating that this strain might be a helpful vector for cystic fibrosis gene therapy or alternative applications [11].

However, among all of the benefits of using bactofection, the risk of infection due to the reversion of virulence in attenuated strains is still a possibility that needs to be addressed. Moreover, the induction of undesirable immune responses is also a possibility that requires careful consideration. Diverse mechanisms have been implemented to mitigate these unwanted risks, such as random chemical mutagenesis (e.g., nitrosoguanidine) to introduce attenuating mutations in several vital genes. These mutations can affect enzymes of crucial metabolic pathways causing auxotrophy, downregulate the surface antigens synthesis to avoid clearance by the immune system of the host, disable the sigma factor of the stress response, and cause the complete inactivation of virulence factors with specific attention to superantigens. The simultaneous attenuation of several genes of vital metabolic pathways may reduce the probability of a reversion to virulence. The use of suicide genes is another commonly used method to reduce the risks associated with bacterial vectors [8,9,12,13].

Suicide genes are genetic elements designed to cause bacterial self-destruction after they have delivered their cargo. One approach that has been used involves the introduction of a gene called LyE (Figure 2). The LyE gene (from phage PhiX174) encodes the enzyme autolysin. When the LyE gene is placed in a prokaryotic expression cassette under the control of an inducible promoter, autolysin can lyse the bacterial wall and release the desired load under appropriate circumstances, as is the case of a construct that employs an inducible promoter activated only in an acidic environment (e.g., within the late macrophage phagosome) [14,15].

## 2. Bacterial Vectors Used as Vaccine Candidates

### 2.1. Main Efforts Using Bacterial-Based Experimental Vaccines

Bacterial-mediated gene transfer for vaccination purposes possesses unique properties that confer advantages over traditional platforms, such as tissue tropism, cell-to-cell spread, virulence factors that mediate the delivery of the codified antigen to inaccessible tissue layers, mediation of the transfer of large genomic fragments, the ability of the vector to act as a potent adjuvant enhancing the immune response against the encoded antigen, and lower costs of production [16,17,18]. Table 1 describes the main efforts so far in producing live attenuated bacterial-based vaccine candidates.

The percentage of use of each type of strain for vaccination is illustrated in Figure 3.

The advantages mentioned above are of particular interest in therapy against diverse infectious diseases, as in the case of designing effective vaccine candidates against HIV, in which Karpenko and colleagues tested in mice an orally administered *Salmonella enteriditis* vector harboring genes codifying immunodominant HIV antigens in a promising strategy [19]. In this study, a eukaryotic expression construct was designed to encode a polyepitope protein TCI (T-cell immunogen), which comprises over 80 cytotoxic T-lymphocyte epitopes from HIV-1 subtype A, B, and C. Rectal immunization of BALB/c mice resulted in the appearance of immunoglobulin gamma (IgG) antibodies against TCI that were capable of neutralizing and inactivating HIV-1. In addition, effective T cell activation was demonstrated by the appearance of significant IFN-γ levels. The antibody titer reached its maximum (1:700) on day 35 after the first immunization (day 7 after the second immunization) and then decreased over 28 days to a level of 1:470. After double immunization, the vector was undetectable in the spleen, liver, and lungs, demonstrating that both the vector and the immunization strategy used in this study can be effective in inducing both humoral and innate immunity and viral clearance of HIV [19].

Vaccination based on virus-like particles (VLPs), despite being safe and immunogenic, presents serious difficulties in their production and is not completely effective in inducing a consistent immune response. In this sense, bacterial vectors may be helpful to synthesize and secrete the required antigenic proteins that will later self-assemble to form functional VLPs. Another important benefit of using bacterial vectors is their inherent capacity to act as a potent adjuvant that may potentiate the mucosal immune response. This hypothesis was proven by Li and colleagues, who demonstrated in a murine study that the *S. choleraesuis* vector was able to synthetize and excrete the capsid protein of the porcine circovirus type 2 (PCV2) with its consequent self-assembly, finally producing functional VLPs, those that induced a capsid specific Th1 immune response and neutralizing antibodies (NAbs) against PCV2. Virus genome copies were significantly lower in treated mice compared with the control group. This study provides an alternative to reduce the difficulties of VLPs production and enhance their immunogenicity, demonstrating that bactofection can be used to increase the effectiveness of other vaccination platforms [20].

In the attempt to control infectious diseases, the most commonly employed strain in bacterial-based vaccination is *S. typhimurium*, due to its inherent infective characteristics and the extensive knowledge of its biology. Regarding vaccination efforts against the COVID-19 pandemic using bactofection, Yoon and colleagues published a study demonstrating the feasibility of inducing effective cellular and humoral immunity by the oral administration of a transformed *S. typhimurium* in mice. This approach employed the *Salmonella* TTSS to secrete the spike antigen into the macrophage cytoplasm. A strong immune response was demonstrated by the detection of anti-spike IgG NAbs (1:700), an increased activity of cytolytic T lymphocyte (CTL) (above 70%), and a significant rise of interferon-gamma (INF-γ) levels [5,32].

Regarding other efforts to develop a SARS-CoV-2 vaccine based on the *S. typhimurium* vector, Jawalagatti and colleagues developed the first multicistronic SARS-CoV-2 vaccine candidate. This experimental vaccine was based on a eukaryotic expression construct containing the *RBD*, *HR*, *M* and *nsp13* genes. When transcribed and translated, these genes produced a fusion protein. The fusion protein was cleaved by the self-cleaving peptide P2A into different peptides, avoiding immunodominant epitope overlapping. This method resulted in an efficient immune response against each one of the coded antigens [21]. P2A is a viral self-cleaving peptide commonly used to separate large polyproteins coded in one ORF into individual proteins. This process is mediated by the interaction of 2A peptide with the ribosomal exit tunnel to dictate an unusual stop of translation (codon-independent termination) at the final Pro codon of 2A [33].

The intramuscular administration of the multicistronic experimental vaccine in mice induced a strong cellular response measured by peptide-specific CD4+ and CD8+ activation and significant changes in the cytokine expression profile of IFN-γ, tumor necrosis factor-alpha (TNF-α), interleukin 4 (IL-4) and IL-10, as well as a robust IgG1 and IgG2a antibody response against RBD, HR, M and nsp13 antigens. Notably, the production of NAbs against HR was significantly more predominant than the rest of the antigens (*p* < 0.01 for IgG1 and *p* < 0.001 for IgG2a). This study established the precedent that a multicistronic intramuscular vaccine against SARS-CoV-2 using the *S. typhimurium* vector is a viable approach [21].

In another attempt to generate an effective *S. typhimurium*-based vaccine candidate against SARS-CoV-2, Zhu and colleagues demonstrated the feasibility of immunization of Wistar rats against spike antigens with just three weekly boosts, detecting antigen-specific IgG NAbs after 28 days post-immunization. Oral vaccination with an attenuated *S. typhimurium* vector may be a viable option to immunize against the desired antigen [22,34].

Mentioning other immunization efforts through bactofection to control chronic infectious diseases, the *Listeria monocytogenes*-mediated gene transfer of IL-12 to control a *Leishmania* major infection was successfully achieved in a murine model. This approach prompted a strong Th1 response and suppressed the Th2 response, showing its potential not only to control a specific infectious disease, but also to provide protection against an allergic response [23].

The world’s deadliest bacterial infection is tuberculosis (TB), whose etiological agent is *Mycobacterium tuberculosis*. This disease causes more than 1.4 million deaths per year, ranking higher than HIV or malaria. Rifampicin is the main standard of treatment against TB, and currently, over a half million cases of rifampicin-resistant TB are being reported annually. The lack of effective therapeutic treatments against TB and its increasing bacterial resistance against antibiotics make the need for an effective vaccine an urgent requirement [35,36,37].

The only licensed vaccine for TB is bacille Calmette–Guérin (BCG), which protects against extrapulmonary TB but not against pulmonary TB. BCG was genetically modified at the Max Planck Institute for Infection Biology to improve its immunogenicity by producing VPM1002. This recombinant BCG-∆ureC::hly live attenuated vaccine candidate has the urease C gene (*ureC*) deleted and replaced by the listeriolysin O (*LLO*) gene from *L. monocytogenes*. Mycobacterial survival in the phagolysosome is promoted by urease through the prevention of acidification. However, if replaced by the LLO gene, acidification occurs, leading to the disruption of the phagolysosomal membrane by LLO. This releases mycobacterial antigens into the host’s cytosol, inducing apoptosis, autophagy, and inflammasome activation in macrophages [24,38,39]. VPM1002 has passed phase I clinical trials in Germany (NCT00749034) and South Africa (NCT01113281), phase II clinical trials in South Africa (NCT02391415 and NCT01479972) and phase III clinical trials in Canada (NCT04439045) and Germany (NCT04435379) and one ongoing trial in India (NCT03152903) [40,41,42,43].

Attenuated strains of *Salmonella* are attractive vaccine candidates for eliciting mucosal and systemic immune responses. In a promising strategy to develop a vaccine against the human papillomavirus type 16, a recombinant *S. typhimurium* strain (capable of secreting the HPV16-L1 protein) was orally and nasally administered to BALB/c mice. The secreted L1 protein was able to form VLPs that closely resemble natural HPV virions as was observed by electron microscopy. This study demonstrated for the first time the capacity of *Salmonella* to be nasally and orally administered and subsequently produce antigen-encoded VLPs capable of inducing a consistent immune response. This was demonstrated by the detection of anti-L1 IgG NAbs capable of neutralizing HPV16 virions in vitro. Anti-VLP IgA and IgG were also detected in oral and vaginal secretions, indicating that potentially protective antibody responses could be induced in mucosal sites, and these could confer protection against HPV and ultimately avoid the cancer risk [25].

*Salmonella enterica* serovar Typhi strain Ty21a is a licensed live-attenuated oral vaccine for typhoid fever that has demonstrated genetic stability for over 30 years [12] and has an optimal efficacy of up to 96% that lasts for over 8 years [44]. Because of this, Ty21a is an optimal vaccine platform for expressing and delivering diverse foreign antigens that can stimulate consistent protection [21,45,46,47]. This protection has been previously demonstrated for Shigellosis [26,27,28], anthrax [29], plague [45], and HPV models [30,31].

Shigellosis remains a major public health problem around the world, causing 212,000 deaths per year. Currently, no licensed vaccines for this disease are widely available, but several vaccine candidates are in development [48,49]. In an attempt to generate a live attenuated vaccine against shigellosis using a recombinant *S. typhi* Ty21a expressing the IO polysaccharide antigen of *Shigella sonnei*, Black and colleagues tested three doses (1–8 × 10^9^ CFU each) of a bivalent experimental vaccine in young adults who, along with unvaccinated controls, were challenged one month later with pathogenic *S. sonnei*. The level of protection conferred by this experimental vaccine against diarrhea was 53%, and 71% against hematest-positive diarrhea. Serum and intestinal immune responses were found, and the presence of IO polysaccharide-specific antibodies IgA and IgG were correlated with protection from illness [50]. Another promising effort in producing a bivalent vaccine candidate using the *S. typhi* Ty21a vector expressing *S. sonnei* LPS resulted in 100% protection against febrile illness (*p* = 0.05) and diarrhea (*p* = 0.04) [26].

### 2.2. Deterioration of Immunity Due to Previous Infections with the Bacterial Vector

A main concern of bactofection as a vaccination platform is the possible impact of preexisting immunity to the employed bacterial vectors due to previous infections or previous vaccinations that could severely affect the immunogenicity of first-time vaccination or boosts that try to augment the immune response. In this context, Metzger and colleagues evaluated the effect on the immune response of a first-time immunization (complete vaccination scheme) with a non-recombinant Ty21a vaccine in 13 volunteers, followed by subsequent administration of a recombinant Ty21a expressing subunits A and B of *Helicobacter pylori* urease (three boosts). Interestingly, the proportion of cellular response to *H. pylori* urease between the non-vaccinated and vaccinated groups did not differ (56% vs. 56%), and urease A and B expression was similar in both groups. These results show that primed individuals may display the same immune response as unprimed individuals [46].

In accordance with this, two independent studies showed that previous priming with a *Salmonella* vector enhanced the response to foreign antigens [51,52]. However, in contrast to this, other studies demonstrated that previous priming with the specific bacterial vector leads to a suppressive effect on the response to codified heterologous antigens. However, this could be overcome by changing the serotype of the desired strain [53,54]. Vector-priming could cause the deterioration of the immune response in a second challenge, but this requires further confirmation. Several attempts to validate this in different clinical trials using Ty21a resulted in inconsistent findings [46].

Major studies should be conducted in endemic zones with a high prevalence of certain diseases to evaluate if priming with a related bacterial strain may deteriorate the immune response to a codified heterologous antigen in the same strain. In an attempt to prove this, a trial was assessed in Indonesia (endemic zone of intense prevalence of typhoid fever) in 23,543 individuals, with the objective of analyzing the efficacy of the complete vaccination scheme of Ty21a to prevent typhoid fever. A 30-month follow-up was carried out, detecting just a 53% level of protection against the disease (*p* < 0.0001). This information contrasts with another similar study in Egypt (endemic region of moderate prevalence of typhoid fever), which showed a protective efficacy of 96% [53].

According to these results, prior priming with the vector may result in the inefficient induction of immunity against codified antigens. To overcome this issue, different serotypes or strains may be used [54,55]. Several attempts, but with inconsistent results, have been made to evaluate whether previous priming of a certain vector may hinder achieving immunity against its harbored recombinant antigens. Some parameters, such as dose, route of administration, and bacterial invasiveness, may be standardized to compare studies and determine a conclusion effectively.

### 2.3. Induction of the Immunity in the Presence or Absence of Phagosomal Escape Genes

Transformed auxotrophic mutants harboring the LLO gene can escape from phagosomes into the cytosol of macrophages and deliver their antigen-encoded eucaryotic expression plasmids after being lysed because of their attenuation [1,11,23,56]. It has been demonstrated that foreign DNA introduced into eukaryotic cells using bacterial vectors with the LLO gene reaches the host cell’s nucleus (the precise mechanism is unknown), with the consequent transcription and translation process and induces an efficient and robust immune response against the plasmid-encoded antigen [1,57]. Moreover, in vivo experiments demonstrated that the transfer of eukaryotic expression plasmids using the *S. typhimurium* vector with truncated ActA and LLO genes could induce an effective immune response against the DNA-encoded antigen, even with a single oral immunization [7].

Other mechanisms of phagosomal escape have been successfully employed in bactofection as the IpaC–IpaB translocon at the tip of the T3SS of *Shigella* [6,9,58]. In contrast, a lower efficiency in the induction of immunity is expected when a strain without a phagosomal escape gene is employed. Several observations confirmed that the transfer of plasmids from intracellular pathogens such as *Salmonella* would be harder to achieve because they mainly remain inside the phagocytic vacuole [6,7,9]. Further studies comparing the efficacy between vectors with or without phagocytic escape genes are required, not only in terms of plasmid release but also in the antibody response, CD8+ and CD4+ activation, and, of course, animal survival.

The efficiency of plasmid expression without a phagosomal escape gene has been evaluated in vitro in two attenuated CD12 and WT05 *S. typhimurium* vaccine candidates. In these experiments the EGFP (enhanced green fluorescent protein) was used. The results indicated that 98% of the J774.2 murine macrophages exposed to the bacterial vector successfully expressed EGFP [59]. Similarly, Darji et al., working with *S. typhimurium* harboring a eukaryotic expression plasmid without a phagosomal escape gene and murine primary peritoneal macrophages, obtained 30% of the cells expressing the reporter protein [7].

It should be noted that there are important differences in the experimental models of these two approaches, like the kind of cell line used (J774.2 vs. primary peritoneal macrophages), or the kind of bacterial strain employed (CD12, WT05, and SL7207). In the different conditions the reporter gene expression was as follows J774.2/WT05 98.16%, J774.2/CD12 74.12%, and J774.2/SL7207 94.54%; primary macrophages/WT05 64%; primary macrophages/CD12 32%; primary macrophages/SL7207 28%. These results lead us to consider that both the cell type used as the model and the chosen bacterial strains can generate important variations in the in vitro approaches; therefore, in vivo studies are essential to diminish biases [58]. However, the authors of these two studies agreed that the level of gene transfer is acceptable, despite low plasmid stability (11.6–13.8%) in the LLO absent constructs [7,59].

## 3. Bacterial-Mediated Cancer Therapy

### 3.1. Main Efforts to Treat Cancer with Bacterial Vectors

The first report of treating cancer with bacteria goes back to 1891, when William B. Coley injected *Streptococcus* into a patient with cancer, reporting the successful shrinking of the malignant tumor. Coley reported more than 1000 tumor patients effectively treated with bacteria or bacterial-related products (Coley’s Toxins). However, his work did not prosper due to criticism and growing disinterest caused by the recent appearance of radiotherapy and chemotherapy, yet modern immunology showed that Coley’s principles were correct and various bacteria can effectively infect cancer cells and limit their growth [60,61]. A relevant aspect to consider is that the bacterial anti-tumor effectiveness in vivo depends on the chosen bacterial species, their genetic background, and their infectious behavior within the tumor microenvironment [62].

Van Pijkeren and colleagues demonstrated for the first time that an attenuated (by increased sensitivity to ampicillin) *Listeria monocytogenes* vector can effectively invade and spread in tumors and deliver genes in vivo in murine models, and in human breast tumor samples ex vivo. They used a firefly luciferase gene as a reporter with expression confined to mammalian cells. Since this strain was highly sensitive to ampicillin, treatment with this antibiotic caused a greater control of lysis and delivery of the transgene both in vivo and in vitro, maintaining a stable expression of the reporter gene for more than a week. This study set the precedent that *Listeria*-mediated gene delivery for the treatment of tumors is a realistic concept [56].

Several authors have reported success in reducing the volume of solid tumors and increased survival in mice by infecting them with bacteria, especially when using the *S. typhimurium* vector, which has displayed impressive and consistent results, mainly explained by its inherent characteristics, such as its capacity to survive in acidic and hypoxic environments [63]; virulence factors encoded in its pathogenic Island I and II (SPI-I/II) that enable the infection of tumor cells [64]; transmembrane receptors such as TAR and TRG that facilitate its migration toward tumor environments [65]; and other genes like motility genes (e.g., *motAB*), chemotaxis genes (e.g., *cheY*) and ethanolamine ammonia-lyase light chain (*eutC*) that code for proteins that enhance tumor colonization. All of these properties make *S. typhimurium* an ideal vector for cancer treatment [66,67].

One additional approach to antitumor therapy using bacterial vectors is gene-directed enzyme prodrug therapy (GDEPT). This method involves delivering a gene-coding enzyme to specific cells, which then controls the conversion of prodrugs into drugs. This process increases toxicity and induces apoptosis in the target cells [68].

Following this strategy, a study aimed to evaluate the anticancer efficacy of orally administrated *S. typhimurium* vector, transformed with an *E. coli* purine nucleoside phosphorylase (ePNP) gene. The killing effect or apoptosis induction in murine melanomas and pulmonary tumors was evaluated, as well as the decrease in tumor volume and mouse survival. ePNP phosphorylase catalyzes the interconversion between 6-methylpurine 2′-deoxyriboside (MePdR) to 6-methylpurine. This therapy was successful, achieving an effective tumor volume reduction of 59–80% when the tumor was >500 mm^3^, and a complete tumor regression and long-term cure when the tumor size was less than 300 mm^3^, proving that this *S. typhimurium* vector harboring ePNP may be helpful in cancer therapy [66].

The most common bacterial vector used in cancer therapy is *S. typhimurium*, which has produced different grades of success, mainly because of differences in the codified therapeutic genes and slight differences in attenuation mechanisms implemented in each study. Some examples are lung cancer [69,70], melanoma [71,72], colon cancer [73,74,75,76,77,78], prostate cancer [79,80,81], B-cell lymphoma [82,83], T-cell lymphoma [84], breast cancer [69,85,86,87,88], osteosarcoma [89,90], pancreatic cancer [91,92], and glioma [93].

Other strains used as bacterial vectors for cancer therapy are *Bifidobacterium longum*, *B. infantis*, *S. choleraesuis*, and *Toxoplasma gondii* for melanoma and solid tumors [94,95,96,97], *B. longum* and *B. adolescentis* for liver cancer [98,99], *Clostridium butyricum* for glioma [100], *E. coli* for colon cancer [101], and *Lactobacillus rhamnosus* for bladder cancer [102], among others. The list of the above-mentioned studies is displayed in Table 2 and the percentage of use of each type of strain is illustrated in Figure 4.

One of the main antitumor mechanisms of bacterial-based therapy is the induction of cytotoxicity by the bacterial vector, leading to the inhibition of tumor growth and triggering tumor cell death by apoptosis or pyroptosis. In addition, the detection of bacterial pathogen-associated molecular patterns (PAMPs) by immune cells induces cytokine release and recruitment of leukocytes, which initiate an anti-tumor immune response. Moreover, the transfer of antigens from the bacteria to the cancer cell enables the adaptive immune system to recognize the cancer cell as infected and a bearer of exogenous antigens [104,105,106].

Bacterial internalization and intra-cellular replication inside cancer cells are mediated mostly by Type III secretion systems (TTSS). Using this type of secretion system, *S. typhimurium* can introduce bacterial factors that allow its internalization and posterior induction of cell stress responses through the recognition of danger-associated molecular patterns (DAMPs), which in collaboration with bacterial pathogen-associated molecular patterns (PAMPs) activates the immune system, triggering cytokine release and recruitment of leukocytes capable of initiating anti-tumor immune responses. There are direct links between the direct toxicity induced by the bacteria itself and indirect tumor cell death triggered by the immune system [104,105,107].

The use of antitumoral bacterial therapy in tumor-associated macrophages is particularly interesting because it can trigger pyroptosis, a type of pro-inflammatory programmed cell death. This leads to the activation of caspase 1, which in turn activates the inflammasome, triggers the secretion of IL-1B and IL-18, causes changes in cell shape, reorganizes the cytoskeleton and nucleus, and ultimately leads to the rupture of the cell membrane. This results in the release of inflammatory signals that boost the body’s immune response against tumors. This quick cell death of macrophages leads to the liberation of bacteria that can infect surrounding cancer cells. Malignant cells that survive infection may degrade bacterial proteins (by proteasomal degradation) and present foreign peptides through MHC I to cytotoxic lymphocytes, which in turn induce tumor cell death, consequently leading to a rapid decrease in tumor volume [107,108,109,110,111,112].

Another added benefit of treating tumoral tissues with *Salmonella*-related vectors is the prevention of angiogenesis, explained by the downregulation of hypoxia-inducible factor 1-alpha (HIF-1α) and the vascular endothelial growth factor (VEGF). The *Salmonella* effector proteins (codified in their SP-I and II) probably interact with HIF-1α and with its upstream signal mediator protein kinase B (AKT), and in this way downregulate their expression. This specifically inhibits tumor angiogenesis by downregulating the AKT/mTOR pathway [113]. *Salmonella* infection also decreases VEGF expression in tumor cells by reducing AKT phosphorylation (some *Salmonella* effector proteins act as phosphatases, e.g., SptP). This angiogenesis-suppressing effect is amplified by cellular proteins such as protein connexin 43 (Cx43), which are activated under these conditions of HIF-1α and VEGF downregulation [114].

In a more recent study, Chen and colleagues, developed a recombinant model of *Staphylococcus epidermidis* that expressed melanoma tumor antigens anchored to cell-surface or secreted proteins. Upon colonization in mice, the engineered *S. epidermis* elicited tumor specific T-cells that infiltrated local and metastatic lesions exerting cytotoxic activity, causing a significant reduction in tumor volume. This study demonstrated that the immune response to a skin-colonizing bacterium can promote cellular immunity in a region of interest and in distal areas. Furthermore, this therapy can be redirected against other therapeutic targets by expressing the desired tumor-associated antigens [115].

Bacterial-based antitumoral therapy has multiple functionalities. In an interesting approach, Siddiqui and colleagues developed a system in which a tumor colonized with *E. coli* was utilized to deliver specific radiopharmaceutical tumor ligands to solid tumors, and once bound to their cognate receptor, the radioactive nuclides emitted cytotoxic doses of *α* and *β* particles to the surrounding cancer cells, resulting in significant attenuation of the tumor growth and the survival extension of mice. This approach was also correlated with promising anti-tumor immunity, with a noticeable CD8+ T:Treg cell ratio [116]. The use of bacterial biofilms to threat solid tumors is another interesting and novel approach in which the infection of the tumor tissue elicits a host response characterized by a strong neutrophilic influx, which in consequence can induce an indirect antitumoral response [117]. In counterpart, other studies have claimed that biofilms might affect cancer biology by modulating the metabolome, yielding metabolites that enhance cancer growth with special associations with right-sided colon adenomas, so their true efficacy must be elucidated before implementing them in clinical trials [118].

In a more conventional approach, a recombinant *Salmonella* was engineered to express the therapeutic agents IL-2 and TRAIL (known for their antitumorigenic roles) inside colonized tumors, demonstrating an effective cytotoxic activity and conferring an increase in survival, showing again that this is a promising vector for use in cancer therapy [119]. The IL-2 strategy has progressed to a phase II clinical trial (NCT04589234), treating metastatic pancreatic cancer showing favorable preliminary results [120].

### 3.2. Clinical Trials Treating Cancer with Bacterial Vectors

Only 13 studies in Phase I and just two in Phase II have been carried out to date to treat cancer patients with bacterial vectors (Figure 5).

Schmitz-Winnenthal et al. developed the first orally applied tumor vaccine candidate, VXM01. They used a live attenuated *Salmonella typhi* carrying an expression plasmid encoding the VEGFR2 (receptor for the vascular endothelial growth factor A) to induce an anti-angiogenic effect. Patients with an advanced stage of pancreatic cancer were treated with this experimental vaccine in a phase I study. This trial demonstrated an effective anti-angiogenic response mediated by T-cell activity after four priming vaccinations. Specific EGFR2-T cell responses measured by IFNγ showed a positive correlation between the dose, boosts, and IFNγ level, being >1.8-fold higher in treated patients than in the placebo group. The most frequent adverse events were mild, including a decrease in lymphocyte count (22% vs. 0%), an increase in neutrophil count (16.7% vs. 0%), a decrease in platelets (44.4% vs. 12.5%), diarrhea (22% vs. 0%), and nausea (16.7% vs. 0%) [121]. Safety and a consistent immune response were achieved; nevertheless, to validate this therapy, other studies with a greater number of participants and measurements of other variables, such as survival and reduction of tumor volume, among others, are required.

In another trial, assessing the efficacy of tumor regression by the intravenous administration of an attenuated *S. typhimurium* to 24 patients with metastatic melanoma and one patient with metastatic renal cell carcinoma, significative proinflammatory cytokine levels were found (IL-1β, IL-6, IL-12, and TNF-α). This is very interesting because hypothetically, this could revert the characteristic tumor immuno-suppressive microenvironment causing tumor regression. This specificity of infecting tumor cells may be due to the implemented attenuations in *S. typhimurium*, in which the deletion of the *purI* gene causes purine auxotrophy. Since tumor cells are a rich source of purines, they could act as a chemoattractant for *Salmonella*, thus inhibiting tumor growth. On the other hand, *msbB* deletion reduced the toxicity associated with lipopolysaccharide (LPS) by preventing the addition of a terminal myristyl group to the lipid A domain, causing a markedly diminished capacity to induce TNF-α, conferring greater tolerability to the vector. Effective tumor colonization and an intra-tumor administration route have also been successfully demonstrated [122].

Other strategies, such as converting prodrugs into chemotherapeutic drugs at specific sites of tumor proliferation, have been explored. This strategy aimed to avoid the characteristic systemic collateral damage of these drugs. The *E. coli* cytosine deaminase gene has been cloned in a *Salmonella* vector that was administered intratumorally to refractory cancer patients with the purpose of generating the conversion of the prodrug 5-fluorocytosine (5-FC) to 5-fluorouracil (5-FU) just in the tumor tissue, which was effectively achieved. This study demonstrated that *Salmonella* can deliver the functional cytosine deaminase gene to malignant tissues and cause the conversion of prodrugs at specific required sites [123]. Table 3 displays ongoing or completed clinical trials seeking to use *Salmonella* as a cancer treatment.

## 4. Vaccination Based on Bacterial Ghosts

### 4.1. Effectiveness of Bacterial Ghosts Loaded with Therapeutic Nucleic Acids or Drugs

Bacterial ghosts (BGs) are fully inactivated bacteria composed of bacterial shells with pores, with a partial or complete release of their cellular components. If an appropriate inactivation method is used, the structural integrity of the surface antigens can remain intact [124,125,126,127]. BGs can act as potent adjuvants due to their inherent capacity to induce a very strong and effective humoral and cellular response [127]. Because of their highly conserved bacterial cell wall components (lipopolysaccharides, lipoproteins, peptidoglycans, flagellum, fimbriae, pili, and adhesins, among others), which can act as PAMPs [124] and be recognized by pattern recognition receptors (PRR) on immune cells [128], they can stimulate the maturation and differentiation of antigen-presenting cells (APCs), resulting in cytokine secretion (e.g., interleukin-12) and the activation of a specific immune response (e.g., Th1) [129]. This process favors a potent and effective immune response to the encoded heterologous antigens [130].

Gram-negative bacterial inactivation mediated by the inducible expression of the lysis E protein (LyE) from phage ΦX174 is the most effective non-denaturing method employed. This approach allows for the maintenance of the structural integrity of the cell wall antigens, which could potentiate the immune response in favor of the codified antigens. LyE works mainly by two mechanisms. Firstly, it directly inhibits the phospho-MurNAc-pentapeptide translocase (MraY) enzyme, which catalyzes the cell wall peptidoglycan layer [131]. Secondly, the hydrophobic N-terminal of LyE binds to the inner membrane, causing a conformational change that permits the binding of the hydrophobic C-terminal to the outer membrane of the cell wall (across the inner and periplasmic spaces), leading to the formation of 40–200 nm tunnels through which the cytoplasmic content is expelled [132]. The lytic efficiency of LyE reaches 99.99%. However, to kill potential non-lysed bacteria, ghost preparations are inactivated with antibiotics (e.g., gentamycin and streptomycin) or β-propiolactone [15,133].

After at least three washes with PBS, the empty inner space of BGs can be filled with nucleic acids, proteins, enzymes, drugs, or other desired compounds (Figure 6). Additionally, BGs can be closed or sealed with membrane vesicles to preserve their contents [134,135]. These freeze-dried preparations guarantee their stable preservation for months at ambient temperature [132].

BGs can carry more nucleic acids than living bacteria, which makes them ideal as vectors in gene therapy. Their excellent loading capacity with pDNA or dsDNA (up to 6000 medium-sized plasmid copies per BG) is due to the electrostatic interaction between the negatively charged DNA molecules and the positively charged moieties (amine groups) present on the nonpolarized internal membrane [136,137]. The simple DNA loading process is achieved by the resuspension of powdered dried lyophilized BGs in a highly concentrated pDNA solution followed by extensive washing steps. The final amount of DNA present within the BGs depends on the concentration of the DNA solution used [137].

Several other methods of binding DNA to the inner membrane of the BGs have been described [134,138,139]. For more efficient delivery of nucleic acids, condensing the nucleic acids with polymeric polycations inside the ghosts should provide some degree of protection against degradation [134]. After the bacteria has been phagocyted by the APC, it has been demonstrated that eukaryotic expression plasmids reach the nucleus of host cells by an unknown mechanism. Once in the nucleus, they transcribe their encoded product, normally an antigen [1,56].

Several studies in different animal models sought to determine the appropriate dose and route of administration of BGs delivering recombinant DNA encoding antigens of interest to achieve an effective immune response [133,140,141,142,143]. Also, the specific impact of BGs on macrophages [137], DCs [140], tumor cells [144,145], endothelial cells [145] and epithelial cells [146] has been evaluated.

In an approach to determine the expression level of a foreign gene in immune human cells mediated by BG therapy, non-mature human monocyte-derived dendritic cells (DCs) were treated with BGs (*E. coli* vector) harboring a pDNA encoding GFP. The treatment was performed in the presence of a maturation mix (TNF-α, IL-1β, IL-6, and PGE2), resulting in a GFP expression level of 85% in at least 77% of the DCs. Of note was the absence of cytotoxicity induced by the BGs. These results set a precedent for future preclinical trials of BGs as carriers in vaccination [133,147]. The above results are consistent with other in vitro studies in murine RAW264.7 macrophages showing an efficient gene transfer and expression of GFP (>60%) without any further stimulation [136]. The expression efficiency of the transgene is maintained, not dependent on the type of vector used; similar patterns of GFP expression were displayed using *Mannheimia haemolytica* ghosts in DCs, also, the efficiency of BGs taken up by APCs is estimated to be between 52–82% [140,148].

The effectiveness of immune response induction mediated by BGs harboring antigenic genes depends on several factors such as dose, route of administration, and inherent characteristics of the antigenic gene construct. In a murine study comparing the intramuscular vaccine efficiency of naked DNA vs. BG (*M. haemolytica*) treatment codifying the β-galactosidase antigen, a significantly more efficient antigen-specific humoral and cellular (CD4+ and CD8+) immune response (*p* = 0.05) was achieved in the BG group, even with a single dose. The BG treatment modulates the response from Th1/Th2 to a more dominant Th2 pattern. Intravenous immunization with dendritic cells loaded ex vivo with β-galactosidase containing ghosts also elicited efficient specific responses. These findings indicate that BGs represent a more effective and promising strategy compared to naked DNA vaccines [140]. Several other studies demonstrated that BG-based DNA vaccine candidates induce stronger humoral immunity, lymphocyte proliferation, and cellular immunity (mainly Th1 polarization) than naked DNA vaccines or BGs alone. This has been shown in the prevention of *Chlamydia psittaciis* infection by *E. coli* ghosts or the prevention of *Neisseria gonorrhoea* infection by *S. enteritidis* ghosts [149,150]. A remarkable benefit of BGs over other platforms is that BG-based DNA vaccine candidates promote greater DC maturation and the upregulation of costimulatory molecules such as CD80, CD86, CD40, and MHC-II, resulting in a better stimulatory effect on humoral and cellular immunity (e.g., IFN-γ levels). Additionally, BG-based vaccine candidates harboring multiple plasmids instead of a single plasmid maintain a strong immunogenicity [151,152].

Intramuscular, oral, mucosal, intradermal, and peritoneal administration of BG-based DNA vaccination, even with a single dose, demonstrated the same relative long-lasting efficacy, within 12 days after immunization, as well as a preference for the Th1 response, with some exceptions [140,141,153]. A notable result is that regardless of the type of strain and animal model used, BGs or DNA-loaded BGs are capable of consistently providing protection against challenges. This applies whether the challenge involves the bacterial strain, or the antigen encoded in the DNA [140,154,155,156,157,158,159]. Even though there are currently no approved vaccines based on this potential vaccination platform, in vitro and in vivo studies show promising results [140,141,142,143,144,145].

The current trend in vaccination is the use of lipid nanoparticles (LNPs) that encapsulate nucleic acids (e.g., BNT162b2 or mRNA-1273) [160,161]. However, despite the rise of this strategy, the potential of ghost bacteria over the LNP vaccination platform lies in their greater capacity for loading genetic material (up to 6000 therapeutic plasmids per ghost), as well as their inherent ability to act as a potent adjuvant that leads to a strong, consistent, effective, and long-lasting immune response [127,136]. The risk of greater adverse effects and the long-term stability of mRNA in ghosts compared to LNPs has not been fully elucidated.

### 4.2. Cancer Therapy Based on Bacterial Ghosts Harboring DNA

To evaluate the effectiveness of BGs as DNA delivery vectors in cancer therapy, the capacity of *E. coli* and *Mannheimia haemolytica* BGs to bind and be phagocyted by eight different human melanoma cell lines (Owes, SK-Mel-28, A-375, 1F6, 1F6m, WM-164, WM-239 and WM-373) was assessed. This resulted in 82% of the cells expressing the plasmid-encoded reporter gene, with a similar pattern observed in non-malignant APCs. BGs are attractive targets for phagocytosis by melanoma cells primarily because of the membrane LPS that activates TLR-4 (constitutively expressed in melanoma cells) enhancing the production of IL-8 and cell adhesion. The immune evasion of tumor cells may be reversed by delivering genes encoding appropriate proteins to address known defects, to restore the antigen presentation, to recover the cell’s native functions, and to restore the immune response against the tumor cell. In addition, another strategy to increase the immunogenicity of tumor cells is encoding cytokine genes (e.g., IL-2) on the same construct to attract T and NK cells capable of eliminating the tumor [144,162].

Effective gene delivery requires relatively low DNA concentrations to produce expression in target cells, allowing BGs to be loaded with multiple plasmids and be used to deliver heterologous genes [144]. Research teams in this area focus their main efforts on delivering plasmids encoding tumor antigens to DCs and tumor cells to determine if this approach could be useful, considering the adjuvant properties of BGs that could affect them positively and lead to an effective Ag-specific T CD8+ cell response [148]. Rabiei and colleagues studied the effects of the heat-inactivated bacterial strains *S. typhi*, *Staphylococcus epidermidis*, *E. coli* and *Pseudomonas aeruginosa* on the cancer cell lines MCF7 and HT-29, concluding that bacterial infections are cancer-deteriorating agents, and each cancerous cell type is especially adversely affected by a certain bacterial strain. Therefore, an efficient treatment against a specific tumor type may not be effective against another, and these specificities must be determined in future studies [163].

## 5. Conclusions and Perspectives

The use of attenuated or inactivated recombinant bacterial vectors to deliver therapeutic or immunogenic genes in cancer gene therapy and vaccination has shown many advantages, but important aspects continue to be unknown. Therefore, many areas remain unexplored, such as finding specific attenuations that confer the most suitable characteristics for invading tumor tissues, the most appropriate strains with specific pathogenicity islands to invade difficult-to-reach tumor regions, target acidic and hypoxic environment characteristics of the tumor microenvironment, and select the most appropriate virulence factors that confer tropism for specific cell types or tissues. Exploring novel phenotypic and genotypic characteristics of the vectors and possible variations of the encoded therapeutic genes are required to assess if the effectiveness of already established models can be enhanced.

Additionally, comparative studies comparing the efficiency between live and inactivated bacteria harboring therapeutic genes are required, as both models confer appropriate immunity levels against the encoded antigen. Something important to note is that of the 27 studies referenced in this review focused on cancer therapy based on bactofection (Table 2), 70% were based on the *Salmonella* vector, and 55.5% of them were specifically based on the *S. typhimurium* vector. In the live bacterial-based vaccine candidates referenced in this review (Table 1), 85.7% used *Salmonella* vectors. Concerning clinical trials (Table 3), the only vector used was *S. typhimurium*. This gives us an overview of how centralized the research is and that *Salmonella* has turned out to be an ideal vector, but it could also result in ruling out other possible vectors that could be suitable for particular models, for which we urge not to neglect the possibility of using novel vectors.

Since there are already highly effective approved vaccines based on live attenuated bacteria such as Ty21a (approved by the U.S. Food and Drug Administration (FDA) and the World Health Organization (WHO), manufactured by Berna Biotech, Bern, Swiss; PaxVax, San Diego, US; Sanofi Pasteur, Lyon, France; Biopharma, Lima, Perú), which can be used as a platform for future vaccines or cancer therapy [12,27,79], there is the need to create recombinant models of Ty21a that can express the protease *GtgE* (from *S. typhimurium*). This protease can cleave GTPase Rab29, enabling the strain to survive inside phagosomes [163] and give the bacteria enough time to deliver the encoded heterologous antigen, thus obtaining a more accurate approximation in a first animal model and setting a precedent for a possible clinical trial. Other interesting approaches in the near future might include combining encoded heterologous antigens in eukaryotic expression vectors with secreted fully ensembled antigens or even combining them with live or inactivated bacteria loaded with therapeutic drugs.

## Figures and Tables

**Figure 1 vaccines-12-00968-f001:**
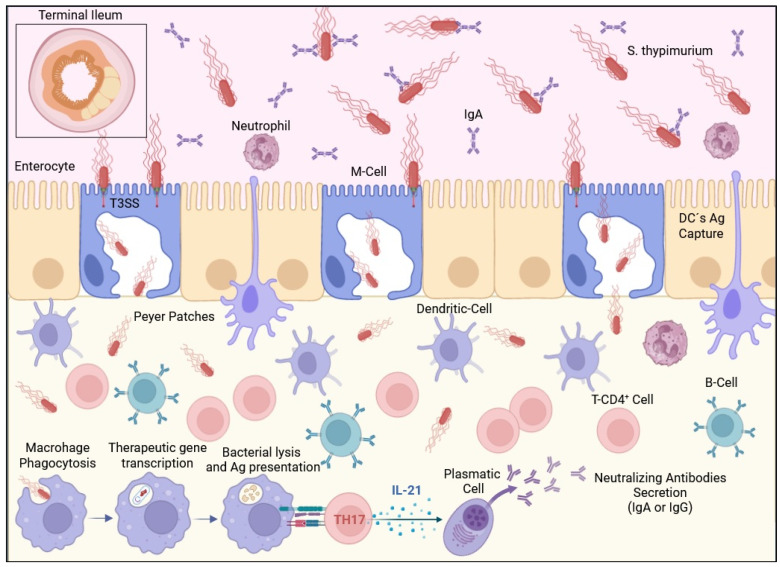
Main mechanisms of bactofection to induce an immune response against a codified antigen. Oral administered *S. typhimurium* (transformed with a certain construct) reaches the terminal ileum and infects M-cells through their type III secretion system (T3SS), then is subsequently translocated to the Peyer patches where macrophages phagocyte it. The acidic environment of the phagolysosome activates an inducible promoter within the construct, leading to the transcription of the desired genes. After bacterial lysis, the mRNA reaches the cytosol, where it is translated into protein; this exogenous protein is then processed and presented in the MHC II context to induce a strong humoral immune response against the desired antigen.

**Figure 2 vaccines-12-00968-f002:**
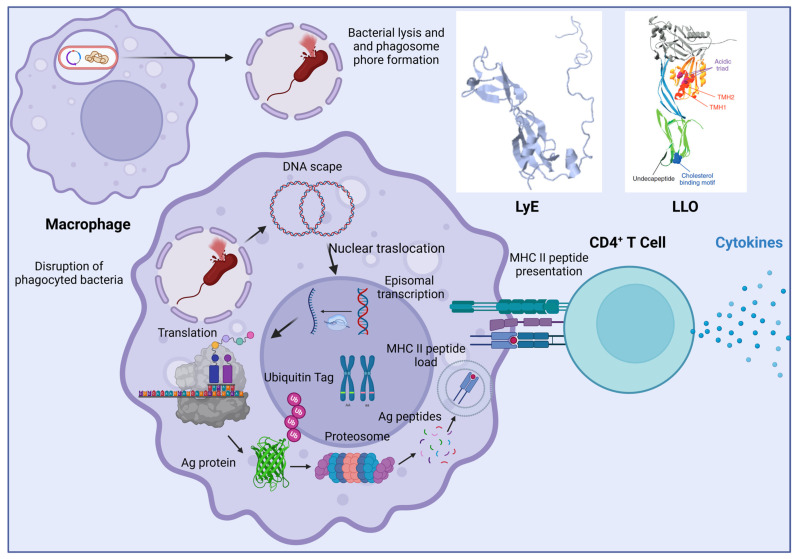
Bacterial and phagolysosome disruption, mRNA release, translation, antigen presentation, and T CD4+ activation mechanisms. In the transformed bacteria, the transcription of LyE and LLO lysins is activated under the phagolysosome’s acidic environment, releasing the cargo mRNA. This is followed by protein translation, proteasomal degradation, MHC II presentation, and subsequent CD4+ T cell activation. LyE transcription triggers bacterial lysis, while LLO transcription induces the formation of pores in the phagolysosome membrane, through which mRNA escapes.

**Figure 3 vaccines-12-00968-f003:**
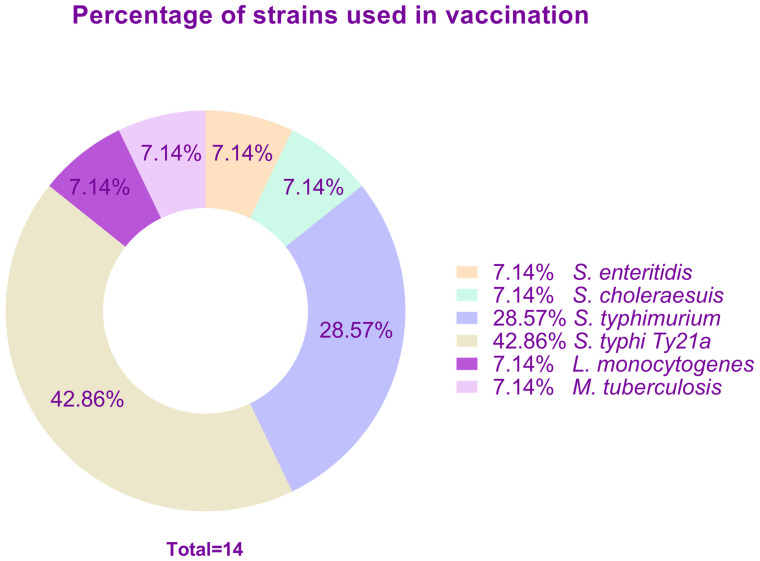
Bacterial strains most commonly used in vaccination by percentage.

**Figure 4 vaccines-12-00968-f004:**
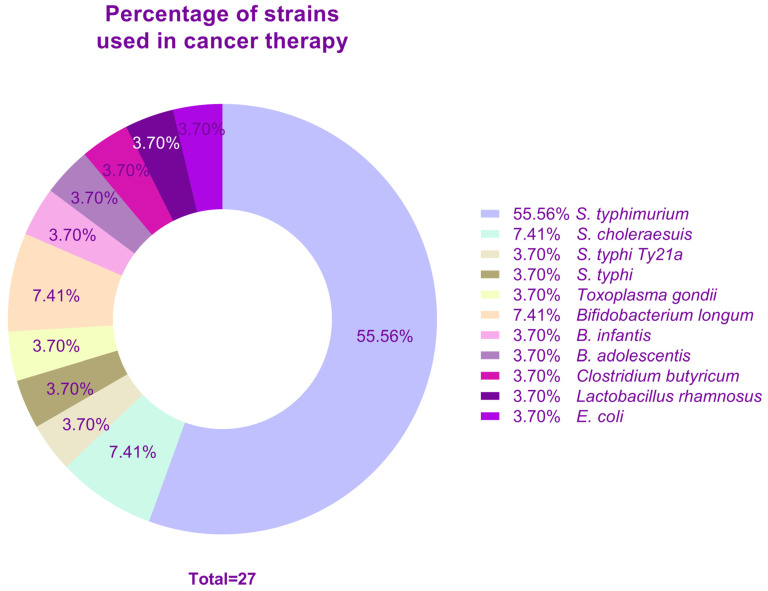
Percentage of bacterial strains most commonly used in cancer therapy.

**Figure 5 vaccines-12-00968-f005:**
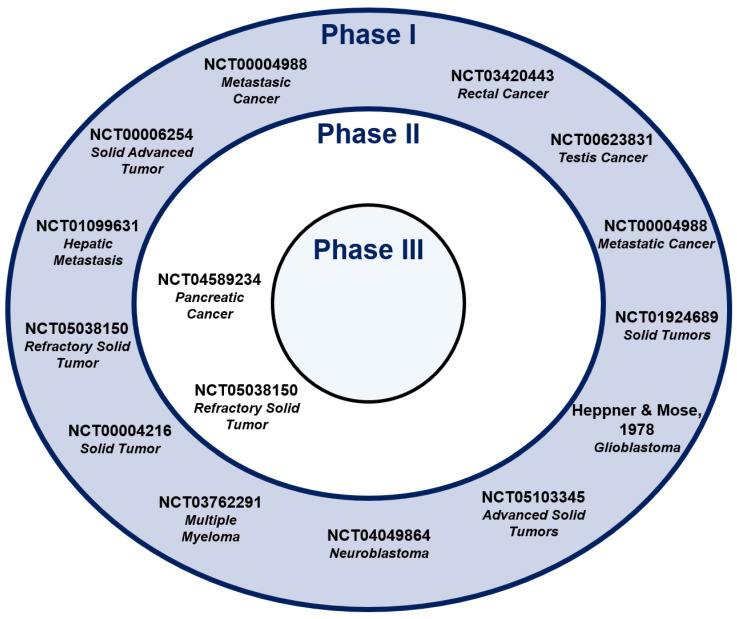
Current level of progress in clinical trials employing bacterial vector-based anticancer treatments. Main clinical trials according to the phase and specific pathological approach.

**Figure 6 vaccines-12-00968-f006:**
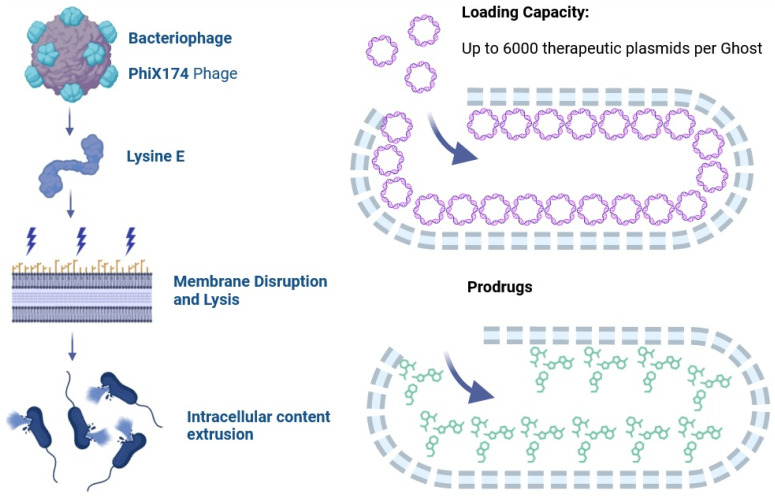
Preparation Method and Loading Strategies for Ghost Vaccine Candidates with Therapeutic DNA or Drugs. LyE protein from phage PhiX174 disrupts bacterial membrane leading to intracellular content extrusion. After several washes with PBS and centrifugation cycles, the ghosts are loaded with the desired therapeutic content. Up to ~6000 therapeutic plasmids per ghost or a not specified amount of chemotherapeutic drug or prodrug is loaded in the ghost bacteria. In both cases, the cargo is not covalently attached to the inner ghost membrane.

**Table 1 vaccines-12-00968-t001:** Main efforts to produce live attenuated bacterial-based vaccine candidates.

Disease	Strain	Encoded Antigen or Strategy	Dosage	Model	Reference
Human immunodeficiency virus (HIV-1)	*S. enteritidis*	Polyepitope protein comprising 80 CTL epitopes from subtype A, B and C HIV-1 proteins	Two rectal doses of 10^8^ CFU with a 4-week interval	Murine	[19]
Porcine circovirus type 2 (PCV2)	*S. choleraesuis*	Cap protein to produce functional VLPs	Two oral doses of 10^9^ CFU with a 3-week interval	Murine	[20]
SARS-CoV-2	*S. typhimurium*	Fusion of SipB160 signal peptide with spike protein to secrete it via TTSS into APC cytosol	Three oral doses of 10^7^ CFU with a 2-week interval	Murine	[5]
SARS-CoV-2	*S. typhimurium*	RBD, HR, M, nsp1, nsp2, nsp3 and nsp4 (nsp13) proteins	Single intramuscular dose of 10^7^ CFU	Murine	[21]
SARS-CoV-2	*S. typhimurium*	Spike protein	Three oral doses of 10^7^ CFU in one-week interval	Murine	[22]
Leishmania	*L. monocytogenes*	IL-12	Single dose of 2 × 10^6^ CFU intraperitoneally or 2 × 10^5^ CFU intravenously	Murine	[23]
*Mycobacterium tuberculosis* TB	*M. tuberculosis*	BCG-∆ureC::hly Deletion of urease C gene and insertion of listeriolysin O from *L. monocytogenes*	Two subcutaneous doses of 10^6^ CFU	Murine	[24]
Human papillomavirus 16 (HPV16)	*S. typhimurium*	L1 major capsid protein inducing the assembly of VLPs	Nasal administration of 5 × 10^7^ at week 0 and 5 × 10^8^ at week 14	Murine	[25]
*Shigella sonnei* Shigellosis	*S. typhi Ty21a*	*S. sonnei* LPS	Three oral doses of 10^9^ CFU within one week	Human	[26]
*S. dysenteriae* Shigellosis	*S. typhi Ty21a*	*S. dysenteriae* LPS	Two doses of intraperitoneal 5 × 10^7^ CFU	Murine	[27]
*Shigella sonnei* Shigellosis	*S. typhi Ty21a*	O polysaccharide (O-Ps)	Single dose of 5 × 10^7^ CFU	Murine	[28]
*Bacillus anthracis* Anthrax	*S. typhi Ty21a*	*PA* gene fused to the secretion signal *hly*	Three intranasal or intraperitoneal doses of 5 × 10^8^ CFU within two weeks	Murine	[29]
HPV16	*S. typhi Ty21a*	L1 major capsid protein inducing the assembly of VLPs	Oral and intranasal single dose of 10^9^ CFU	Murine	[30]
HPV18	*S. typhi Ty21a*	L1 major capsid protein	Single intranasal dose of 5 × 10^8^ CFU	Murine	[31]

**Table 2 vaccines-12-00968-t002:** Main efforts to treat cancer with bacterial vectors.

Type of Tumor	Strain	Encoded Gene or Strategy	Vector	Model	Reference
Lung cancer	*S. typhimurium*	*CCL21* and *survivin*	pBudCE4.1	Murine	[70]
Melanoma	*S. typhimurium*	*6MePdR*	pEZZ-EGFP	Murine	[71]
Melanoma	*S. choleraesuis*	*Thrombospondin-1 (TSP-1)*	pTCYTSP-1	Murine	[95]
Melanoma	*S. typhimurium*	*YS1646-shSTAT3* and *survivin 3342Max*	pWSK29	Murine	[72]
Melanoma	*S. typhimurium*	HSV TK	-	Murine	[103]
Colorectal carcinoma	*S. typhimurium*	*Δ**invG* and *Δ**phoP*	Chromosomal deletion	Murine	[73]
Colorectal carcinoma	*S. typhimurium*	*ΔppGpp*	Chromosomal deletion	Murine	[74]
Breast and colon carcinoma	*S. typhimurium*	*IL-18* & *FasL*	pGEN206	Murine	[69]
Melanoma, colon, breast, and lung carcinoma	*S. typhimurium*	*IL-18* & *Fra-1*	pIRES	Murine	[77]
Neuroblastoma	*S. typhimurium*	*Survivin*	pPRIEG7	Murine	[78]
Prostate cancer	*S. typhi Ty21a*	*siRNA* & *si-HIF-1α*	-	Murine	[80]
B-cell lymphoma	*S. typhimurium*	*aroC mutant*	Chromosomal deletion	Murine	[82]
T-cell lymphoma	*S. typhi*	*guaBA mutant*	Chromosomal deletion	Murine	[84]
Melanoma, breast, and colon carcinoma	*S. typhimurium*	*CPG2*	pTrc99A	Murine	[85]

**Table 3 vaccines-12-00968-t003:** Clinical trials evaluating *Salmonella* as a cancer treatment.

Type of Tumor	Clinical Trial Number	Objective	Phase	Status	Reference
Metastatic pancreatic cancer	NCT04589234	*S. typhimurium* harboring *IL-2* gene will prolong survival and disease progression.	II	Active, not recruiting	https://clinicaltrials.gov/ct2/show/NCT04589234 (accessed on 5 June 2024)
Metastatic cancer	NCT00004988	Stabilize maximum tolerated dose of attenuated *S. typhimurium* in a manner that increases tumor localization.	I	Completed Positive results	[103]
Advanced solid tumors	NCT00006254	Determine the maximum tolerated dose and minimum effective dose of recombinant *S. typhimurium*.	I	Completed	https://www.clinicaltrials.gov/ct2/show/NCT00006254 (accessed on 5 June 2024)
Hepatic metastasis	NCT01099631	Determine effective dose and maximum tolerated dose of *S. typhimurium* harboring *IL-2* gene.	I	Completed	https://www.clinicaltrials.gov/ct2/show/NCT01099631 (accessed on 5 June 2024)
Refractory solid tumors	NCT05038150	Assess safety, tolerability and efficacy of *S. typhimurium* harboring L-methioninase gene.	I and II	Recruiting	https://www.clinicaltrials.gov/ct2/show/NCT05038150 (accessed on 5 June 2024)
Refractory superficial solid tumors	NCT00004216	Determine the maximum tolerated dose, safety and efficacy of *S. typhimurium* in solid tumors.	I	Completed	https://www.clinicaltrials.gov/ct2/show/NCT00004216 (accessed on 5 June 2024)
Advanced solid tumors	NCT05103345	Assess safety, tolerability and efficacy of *S. typhimurium* harboring L-methioninase gene.	I and II	Recruiting	https://clinicaltrials.gov/ct2/show/NCT05103345 (accessed on 5 June 2024)
Multiple myeloma	NCT03762291	Assess safety and tolerability of *S. typhimurium* harboring survivin gene.	I	Active, not recruiting	https://clinicaltrials.gov/ct2/show/NCT03762291 (accessed on 5 June 2024)
Neuroblastoma	NCT04049864	Evaluate the safety and immunogenicity of *S. typhimurium* harboring neuroblastoma-associated antigen and potato virus X coat protein (PVXCP) genes.	I	Unknown	https://clinicaltrials.gov/ct2/show/NCT04049864 (accessed on 5 June 2024)
Refractory solid tumors	NCT01924689	Evaluate safety and efficacy of *Clostridium novyi*-NT spores in patients with treatment-refractory solid tumor malignancies.	I	Completed	https://www.clinicaltrials.gov/study/NCT01924689?cond=NCT01924689%20&rank=1 (accessed on 5 June 2024)
Metastatic cancer	NCT00004988	Evaluate safety and tolerability of *S. typhimurium* VNP20009 with deletions in the *msbB* and *purI* loci in patients with metastatic cancer.	I	Completed	https://www.clinicaltrials.gov/study/NCT00004988?cond=NCT00004988%20%20&rank=1 (accessed on 5 June 2024)
Tumors expressing NY-ESO-1 antigen (cancer-testis Ag)	NCT00623831	Determine the safety and tolerability of a mixed bacterial vaccine candidate that induced a pyrogenic effect in in subjects with malignant tumors that expressed the NY-ESO-1 antigen.	I	Completed	https://www.clinicaltrials.gov/study/NCT00623831?cond=NCT00623831%20%20%20&rank=1 (accessed on 5 June 2024)
Rectal cancer	NCT03420443	Evaluate safety, tolerability and efficacy of *Lactobacillus plantarum* to reduce inflammation and to diminish tissue damage caused by radiation therapy in patients diagnosed with rectal cancer.	II	Completed	https://www.clinicaltrials.gov/study/NCT03420443?cond=NCT03420443%20&rank=1 (accessed on 5 June 2024)

## Data Availability

Data are contained within the article.

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
