# Peer review of "Bactofection, Bacterial-Mediated Vaccination, and Cancer Therapy: Current Applications and Future Perspectives"

_vaccines, 2024, doi:10.3390/vaccines12090968_

Round 1

Reviewer 1 Report

Comments and Suggestions for Authors

It is an interesting and useful review, and it deserves the publication but not in the current form because it contains a significant amount of incorrect information. For example,  the first sentence of the Introduction is the definition of "Bactofection" word.  And it is an impression that the authors invented it because they do not cite any publication. In reality, the first reference in their Reference list contains this word, but the authors did not cite this reference in Introduction. The second paragraph of the Introduction contains the citation of Schaffner and Cols. paper as Reference [2]. But in reality W.Schaffner is the only author of this publication.      

The authors used the word “vaccines” instead of “experimental” or “candidate” vaccines a lot of time but it is reasonable when they write about preparations which has been licensed and approved by official authorities Ministries of Health. But in the most of cases, and especially in conclusion they write about experimental preparations most of which have not been even tested on humans in clinical trials. Such a usage of the word puzzles readers, and it should be corrected.

A significant part of the manuscript is devoted to the bacterial ghosts (BG). But up to now there were just experimental preparations on the base of BG most of which were not officially tested on humans. Therefore It should be also specially stated in this part of the manuscript.

As for expression of bacterial DNA in eukaryotic cells during infection by bacteria, it has been reliably proven for plasmid DNA. Therefore the reviewer offers the authors to check thoroughly whether it had been proven for the whole genomic bacterial DNA.

As for treatment of cancer by “bactofection”, it would be better not to use this word at all because the scientists who real make breakthrough research in this area do not use this word.

As for citation of important publications, the authors of the manuscript did not cite these recent publications devoted to the cancer treatment by bacteria:

1.       https://www.aacr.org/blog/2023/08/18/its-the-little-things-a-role-for-bacteria-in-cancer-treatment/  - (here are cited three recent important publications);  

2.       Hilla Mills, Ronald Acquah, Nova Tang, Luke Cheung, Susanne Klenk, Ronald Glassen, Magali Pirson, Alain Albert, Duong Trinh Hoang, and Thang Nguyen Van.  The Use of Bacteria in Cancer Treatment: A Review from the Perspective of Cellular Microbiology.// Emerg Med Int. 2022; 2022: 8127137. doi: 10.1155/2022/8127137

Finally, the authors stated in the last paragraph of Conclusion that “Since there are already highly effective approved vaccines based on live attenuated bacteria such as Ty21a, which can be used as a platform for future vaccines or cancer therapy [12, 44, 80],…”  But these reference dated 2009, 2007 and 2014 years. And authors did not mention who approves these “vaccines” and what companies produce these “approved” vaccines, because the reviewer could not find these data. So, it should be corrected or additional data should be added. 

Author Response

Dear Reviewer,

We appreciate your helpful suggestions and comments, all of which we have taken into consideration. We have addressed all suggested points accordingly.

Reviewer 1  (Changes suggested by reviewer 1 are highlighted in blue)

It is an interesting and useful review, and it deserves the publication but not in the current form because it contains a significant amount of incorrect information. For example, the first sentence of the Introduction is the definition of "Bactofection" word.  And it is an impression that the authors invented it because they do not cite any publication. In reality, the first reference in their Reference list contains this word, but the authors did not cite this reference in Introduction.

Response: We acknowledge and appreciate your observation. We have made the necessary corrections in the first paragraph of our introduction to ensure that readers are aware of the term 'bactofection' and its common usage in scientific literature.

The second paragraph of the Introduction contains the citation of Schaffner and Cols. paper as Reference [2]. But in reality W.Schaffner is the only author of this publication.

Response: Thanks. Your accurate comment about the authorship of the cited publication has been noted, and we have made the necessary correction to properly reflect the authorship.

The authors used the word “vaccines” instead of “experimental” or “candidate” vaccines a lot of time but it is reasonable when they write about preparations which has been licensed and approved by official authorities Ministries of Health. But in the most of cases, and especially in conclusion they write about experimental preparations most of which have not been even tested on humans in clinical trials. Such a usage of the word puzzles readers, and it should be corrected.

Response: Thank you for your valuable correction, in response, we have corrected all instances of the term "vaccine" in the text that did not refer to those approved by competent health authorities. We have replaced these with the terms you suggested, such as "vaccine candidate" or "experimental vaccine."

A significant part of the manuscript is devoted to the bacterial ghosts (BG). But up to now there were just experimental preparations on the base of BG most of which were not officially tested on humans. Therefore It should be also specially stated in this part of the manuscript.

Response: Thank you for your insightful observation, in response to it, the requested clarification was added in the penultimate paragraph of section 4.1. clarifying that by now there are no approved vaccines based on this type of platform.

As for expression of bacterial DNA in eukaryotic cells during infection by bacteria, it has been reliably proven for plasmid DNA. Therefore the reviewer offers the authors to check thoroughly whether it had been proven for the whole genomic bacterial DNA.

Response: Thank you for your comment; it was clarified that all studies and information presented in this review regarding potential therapeutic strategies using bacterial vectors utilize plasmid DNA, not bacterial genomic DNA. If you find that any paragraph conveys this incorrectly or is confusing, we would appreciate it if you could bring it to our attention so we can make the necessary corrections.

If this confusion arose in the ghost vaccines section, we could clarify that genomic DNA is always absent due to the washing and preparation methodology indicated in the same paragraph.

As for treatment of cancer by “bactofection”, it would be better not to use this word at all because the scientists who real make breakthrough research in this area do not use this word.

Response: Thank you for your valuable suggestion, in response to this, the term "bactofection" was removed from the cancer treatment section and replaced with terms such as “bacterial vectors”, “bacterial-based therapy”, “antitumoral bacterial therapy”, “Salmonella-related vectors”, among others.

As for citation of important publications, the authors of the manuscript did not cite these recent publications devoted to the cancer treatment by bacteria:

  1. https://www.aacr.org/blog/2023/08/18/its-the-little-things-a-role-for-bacteria-in-cancer-treatment/  - (here are cited three recent important publications);  
  2. Hilla Mills, Ronald Acquah, Nova Tang, Luke Cheung, Susanne Klenk, Ronald Glassen, Magali Pirson, Alain Albert, Duong Trinh Hoang, and Thang Nguyen Van.  The Use of Bacteria in Cancer Treatment: A Review from the Perspective of Cellular Microbiology.// Emerg Med Int. 2022; 2022: 8127137. doi: 10.1155/2022/8127137

Response: Thank you for such a valuable contribution. Information about the aforementioned articles has been added to the end of section 3.1, focusing on cancer therapy using bacterial vectors.

Finally, the authors stated in the last paragraph of Conclusion that “Since there are already highly effective approved vaccines based on live attenuated bacteria such as Ty21a, which can be used as a platform for future vaccines or cancer therapy [12, 44, 80],…”  But these reference dated 2009, 2007 and 2014 years. And authors did not mention who approves these “vaccines” and what companies produce these “approved” vaccines, because the reviewer could not find these data. So, it should be corrected or additional data should be added. 

Response: Thank you again for your valuable comment. The requested information, such as health authorities that approved it and companies that manufacture it, was added to the conclusion section.

Reviewer 2 Report

Comments and Suggestions for Authors

Bactofection has shown to be a promising strategy in vaccination, gene therapy, and cancer therapy. This review summarized the general theory of bactofection and its advantages, disadvantages, challenges, and expectations. The authors also described the current scope of bactofection and promising results. In addition, the authors propose potential novel approaches such as creating recombinant models of Ty21a to enable the strain to survive inside phagosomes and giving the bacteria enough time to deliver the encoded heterologous antigen or to combine encoded heterologous antigens in eukaryotic expression vectors with secreted fully ensembled antigens or even combining them with live or inactivated bacteria loaded with therapeutic drugs. This review is very informative. However, the organization of the review can be improved to make it easier to read. I have the following suggestions and comments.

1.      After the introduction section, the authors may consider combining sections 2-9 into two major sections: bacterial vectors used as vaccine and bacterial-mediated cancer therapy. Please note that section 3 is missing.

2. Table 1 summarizes the main trends in producing live attenuated bacterial-based vaccines very well. The authors may consider moving it up to near line 133, making it easier for the readers to follow.

3.      In Table 1, HPV 16 and HPV 18 can be described as (HPV16) and (HPV 18), respectively, to avoid confusion.

4.      In table 1. The references can be cited as reference numbers to be consistent with how other references are cited in the text.

5.      In the bacterial ghost section, the authors may consider including one paragraph to briefly compare the Pros and Cons of this approach over the lipid nanoparticle approach. This will significantly increase the excitement level of this review.

Author Response

Dear Reviewer,

We appreciate your helpful suggestions and comments, all of which we have taken into consideration. We have addressed all suggested points accordingly.

Reviewer 2 ((Changes suggested by reviewer 2 are highlighted in yellow)

Bactofection has shown to be a promising strategy in vaccination, gene therapy, and cancer therapy. This review summarized the general theory of bactofection and its advantages, disadvantages, challenges, and expectations. The authors also described the current scope of bactofection and promising results. In addition, the authors propose potential novel approaches such as creating recombinant models of Ty21a to enable the strain to survive inside phagosomes and giving the bacteria enough time to deliver the encoded heterologous antigen or to combine encoded heterologous antigens in eukaryotic expression vectors with secreted fully ensembled antigens or even combining them with live or inactivated bacteria loaded with therapeutic drugs. This review is very informative. However, the organization of the review can be improved to make it easier to read. I have the following suggestions and comments.

  1. After the introduction section, the authors may consider combining sections 2-9 into two major sections: bacterial vectors used as vaccine and bacterial-mediated cancer therapy. Please note that section 3 is missing.

Response: Thank you for your valuable comment, the numbering of the sections was corrected so that it is consecutive. We also appreciate your comment about including everything in two large sections; in response to this, we group our information into 3 large sections (with their corresponding subsections): vaccination with bacterial vectors, cancer therapy based on bacterial vectors, and ghost vaccines. We considered that the most appropriate thing would be to open a third section dedicated to bacterial ghosts, not only because of the great length of the first two sections but also because, in technical and bibliographical terms, most authors classify it as a different, well-differentiated therapeutic alternative from the live attenuated bacteria strategy. In this sense, we believe that if we add the section on ghost bacteria to section 2, we could detract from the significance of this therapeutic strategy with important differences compared to live attenuated bacteria.

  1. Table 1 summarizes the main trends in producing live attenuated bacterial-based vaccines very well. The authors may consider moving it up to near line 133, making it easier for the readers to follow.

Response: Thank you for your insightful observation. In response to it, Table 1 has been moved to line 133 so that the reader can more easily follow the text.

  1. In Table 1, HPV 16 and HPV 18 can be described as (HPV16) and (HPV 18), respectively, to avoid confusion.

Response: In response to your accurate comment, the modification was made in Table 1 to reference it as you indicated.

  1. In table 1. The references can be cited as reference numbers to be consistent with how other references are cited in the text.

Response: Your observation seems very relevant to us, which is why the references for both Table 1 and Table 2 have been modified.

  1. In the bacterial ghost section, the authors may consider including one paragraph to briefly compare the Pros and Cons of this approach over the lipid nanoparticle approach. This will significantly increase the excitement level of this review.

Response: Thank you for such a valuable contribution. The paragraph with the requested information has been added at the end of the bacterial ghost section.

Round 2

Reviewer 1 Report

Comments and Suggestions for Authors

The authors made practically all changes the Reviewer suggested in the comments and added new paragraphs which explained more exactly some statements. But one sentence in one added paragraphs on the Page 20 (in lines 664-666) is not correct because it stated that "...vaccines based on LNPs require CONSTANT boosters...". In reality most of the vaccines including Hepatitis B vaccine, HPV and others need at least two injections/boosters. As for mRNA coronavirus vaccines, they require not only one booster but injection of modified  mRNA-LNP vaccine coding evolved recombinant protein because of continuously evolving of the coronavirus S protein.  So, this sentence should be deleted or substantially modified. 

And one my previous comment was about my suggested citation of Reference [1} after the first sentence of Introduction showing that authors did not invent the word "Bactofection"  but took it from this reference. Authors did not do it and therefore they create the false impression that they designed this word. 

Round 2

Comments and Suggestions for Authors

The authors made practically all changes the Reviewer suggested in the comments and added new paragraphs which explained more exactly some statements. But one sentence in one added paragraphs on the Page 20 (in lines 664-666) is not correct because it stated that "...vaccines based on LNPs require CONSTANT boosters...". In reality most of the vaccines including Hepatitis B vaccine, HPV and others need at least two injections/boosters. As for mRNA coronavirus vaccines, they require not only one booster but injection of modified  mRNA-LNP vaccine coding evolved recombinant protein because of continuously evolving of the coronavirus S protein.  So, this sentence should be deleted or substantially modified.

Response: Thank you for your valuable comment, in response to this observation, the sentence you indicated was removed from the paragraph.

And one my previous comment was about my suggested citation of Reference [1} after the first sentence of Introduction showing that authors did not invent the word "Bactofection"  but took it from this reference. Authors did not do it and therefore they create the false impression that they designed this word.

Response: Thank you for your comment, reference [1] was added after the first sentence of Introduction as requested.